# Induction of volatile organic compounds in chrysanthemum plants following infection by *Rhizoctonia solani*

**Dariusz Piesik**[1]◐*, **Natalia Miler**[2]◐*, **Grzegorz Lemańczyk**[1]◐, **Alicja Tymoszuk**[2]‡,
**Karol Lisiecki**[1]‡, **Jan Bocianowski**[3]‡, **Krzysztof Krawczyk**[4]‡, **Chris A. Mayhew**[5]‡*

**1** Department of Biology and Plant Protection, Bydgoszcz University of Science and Technology, Bydgoszcz, Poland, **2** Department of Biotechnology, Laboratory of Horticulture, Bydgoszcz University of Science and Technology, Bydgoszcz, Poland, **3** Department of Mathematical and Statistical Methods, Poznań University of Life Sciences, Poznań, Poland, **4** Department of Virology and Bacteriology, Institute of Plant Protection – National Research Institute, Poznań, Poland, **5** Institute for Breath Research, Universität Innsbruck, Innrain, Innsbruck, Austria

◐ These authors contributed equally to this work.
‡ These authors also contributed equally to this work
* Dariusz.Piesik@pbs.edu.pl (DP); Natalia.Miler@pbs.edu.pl (NM); Christopher.Mayhew@uibk.ac.at (CAM)

**Data Availability Statement:** All relevant data are within the manuscript and its Supporting information files.

## Abstract

This study investigated the effects of *Rhizoctonia solani* J.G. Kühn infestation on the volatile organic compound (VOC) emissions and biochemical composition of ten cultivars of chrysanthemum (*Chrysanthemum × morifolium* /Ramat./ Hemsl.) to bring new insights for future disease management strategies and the development of resistant chrysanthemum cultivars. The chrysanthemum plants were propagated vegetatively and cultivated in a greenhouse under semi-controlled conditions. VOCs emitted by the plants were collected using a specialized system and analyzed by gas chromatography/mass spectrometry. Biochemical analyses of the leaves were performed, including the extraction and quantification of chlorophylls, carotenoids, and phenolic compounds. The emission of VOCs varied among the cultivars, with some cultivars producing a wider range of VOCs compared to others. The analysis of the VOC emissions from control plants revealed differences in both their quality and quantity among the tested cultivars. *R. solani* infection influenced the VOC emissions, with different cultivars exhibiting varying responses to the infection. Statistical analyses confirmed the significant effects of cultivar, collection time, and their interaction on the VOCs. Correlation analyses revealed positive relationships between certain pairs of VOCs. The results show significant differences in the biochemical composition among the cultivars, with variations in chlorophyll, carotenoids, and phenolic compounds content. Interestingly, *R. solani* soil and leaf infestation decreased the content of carotenoids in chrysanthemums. Plants subjected to soil infestation were characterized with the highest content of phenolics. This study unveils alterations in the volatile and biochemical responses of chrysanthemum plants to *R. solani* infestation, which can contribute to the development of strategies for disease management and the improvement of chrysanthemum cultivars with enhanced resistance to *R. solani*.

**Funding:** The author(s) received no specific funding for this work.

**Competing interests:** The authors have declared that no competing interests exist.

## Introduction

Plants communicate to each other using airborne signals known as volatile organic compounds (VOCs), playing a crucial role in conveying information to both conspecific and heterospecific plants about the nature of the stress and to plant mutualists and competitors [1,2]. For example, VOCs emitted by herbivore-damaged plants can inform neighboring undamaged plants to enhance their resistance against a possible attack [1] and facilitate host discrimination and host finding in phytophagous insects [3]. VOCs are also involved in regulating various physiological processes in plants, including enzyme activity, growth, respiration, photosynthesis, reactive oxygen species content, dormancy, and plant-to-plant competition [4]. The largest category of plant-released VOCs are terpenes, including isoprene, monoterpenes, and sesquiterpenes. While monoterpenes and sesquiterpenes are recognized as chemical messengers in plant-insect relationships, isoprene is primarily released to mitigate abiotic stress factors. Recent studies have revealed that terpenes also serve as signals for inter-plant communication [2].

Plant volatiles comprise organic compounds derived from the breakdown of secondary metabolites that are emitted by leaves and flowers in response to stress [5–7]. The composition of these volatiles varies among plant species [8,9]. Green leaf volatiles (GLVs) are a crucial subgroup within the biogenic VOCs. They comprise alcohols, aldehydes, and esters with a six-carbon (C6) structure and are emitted by nearly all plant species [10]. GLVs have various effects on plants. They can repel or attract herbivores and their natural predators [11], induce plant defense mechanisms, prime plants for enhanced defense [12], activate abiotic-stress related genes [13] and exhibit direct toxicity against bacteria and fungi [14].

Certain volatiles have been identified as key players in mitigating oxidative stress induced by high light intensity by scavenging reactive oxygen species, stabilizing membranes, and regulating stress responses, with the most important volatile being isoprene [15]. Plants produce a diverse array of VOCs belonging to different chemical classes, such as terpenoids, benzenoids, phenylpropanoids, fatty acid-derived molecules, and minor chemical classes such as nitriles, (ald)oximes, and sulfides [16]. When subjected to stress, most plant species release similar volatiles, including the monoterpenes (E)-β-ocimene and linalool, sesquiterpenes (E,E)-α-farnesene and (E)-β-caryophyllene, and GLVs, which include (Z)-3-hexen-1-ol and (Z)-3-hexenyl acetate [17–22].

The manipulation of volatiles through metabolic engineering holds considerable potential for controlling agricultural pests [23–26]. GLVs that induce systemic resistance against pathogens could be utilized as "green vaccines" in agriculture to defend against impending pathogen attacks [27]. However, our understanding of the mechanisms by which volatiles induce systemic resistance is still in its early stages. It remains uncertain whether a broad application of volatiles would significantly impact plant productivity. Also, the specific interactions between plants and VOCs when attacked by different fungal species have only been explored in a limited number of studies [28,29]. A diverse scope of VOCs, namely terpenes, aromatics, nitrogen-containing compounds, fatty acid derivatives, as well as the volatile phytohormones methyl jasmonate and methyl salicylate are being produced by plants as a result of pathogenic microbial invasion. Based on the timing of the VOC emissions together with their antimicrobial activity, it is assumed that increased VOC production is a result of the plant's defense system against pathogens. Nonetheless, solid evidence supporting this statement is still lacking [28].

Chrysanthemum (*Chrysanthemum × morifolium* /Ramat./ Hemsl.) is known for its decorative qualities and hence is a highly popular ornamental plant worldwide, second only to roses. It can be cultivated as a potted plant or used as cut flowers. This plant belongs to the

*Chrysanthemum* genus of the Asteraceae family and is native to Central-East Asia [30]. The genus encompasses numerous plant species of great importance, cultivated for their ornamental value, as well as their production of valuable secondary metabolites [31]. *Chrysanthemum cinerariaefolium* is a primary source of pyrethroids, which are natural insecticides derived from plants. Other species, such as *Chrysanthemum indicum* and *Chrysanthemum coronarium*, are utilized in Asian cuisine [31,32]. *Chrysanthemum × morifolium*, owing to its unique biochemical properties, has held a significant place in traditional Chinese medicine for centuries [33]. The dried flower heads of this plant are used to prepare an herbal beverage known as "chrysanthemum tea", which is attributed to possessing anti-inflammatory, antibacterial, antiviral, and antifungal properties. The tea is also said to alleviate symptoms of neurological conditions such as headaches, tinnitus, and Parkinsonism [32,34,35]. These beneficiary effects are attributed to the presence of bioactive secondary metabolites such as flavonoids and VOCs [35,36].

The major goal of this study was to investigate the induction of VOCs in *Chrysanthemum × morifolium* (Ramat.) Hemsl. plants of different cultivars following inoculation with fungal pathogen *Rhizoctonia solani*. The aims of the research presented here are: (1) to determine whether different cultivars of chrysanthemum release varying amounts and types of VOCs, resulting in a qualitatively and quantitatively diverse bouquet of odors; (2) to investigate whether the mode of inoculation (e.g., stem leaf infection versus soil inoculation) leads to distinct VOCs and temporal VOC profiles, thereby indicating different defense responses in the plants; and (3) to establish whether the VOCs emitted by chrysanthemum plants have any detrimental effect on the growth of *R. solani* mycelium, and thus potentially act as a defense mechanism against the pathogen. By studying the induction of VOCs in chrysanthemum and their potential effects on *R. solani*, our results provide valuable insights into the plant's defense mechanisms and contribute to the breeding of resistant chrysanthemum cultivars through the analysis of VOC emissions. Moreover, the performed analysis of primary and secondary metabolite composition sheds light on the biochemical responses of chrysanthemum plants to *R. solani* infestation.

## Materials and methods

Following the inoculation of chrysanthemum plants with *R. solani*, VOCs emitted by the plants were sampled at specific times and analyzed using gas chromatography/mass spectrometry (GC/MS). The growth of *R. solani* mycelium was evaluated by measuring mycelial growth rates or other relevant parameters in the presence or absence of VOCs.

### Plant material

Ten cultivars of chrysanthemum (*Chrysanthemum × morifolium* /Ramat./ Hemsl.) were used in this study, namely, 'Ania', 'Beata', 'Brda', 'Kasia', 'Lidka', 'Luczniczka', 'Malgosia', 'Polka', 'Wda', and 'Zofia', all of which were cultivated under greenhouse conditions. These plants were vegetatively propagated via shoot-tip cuttings and rooted in 64-cell propagation trays using a peat: perlite 2:1 substrate mixture for two weeks, and placed under a perforated transparent film tunnel.

After rooting, the cuttings were individually transplanted into 12 cm diameter pots that had been filled with a peat-based substrate for ornamental plants (Gramoflor, Poland). They were then cultivated in a greenhouse from May to July, with watering and fertilization carried out twice per week using Peters Professional General Purpose 20-20-20 NPK fertilizer (Scotts, USA), following the standard procedure for chrysanthemum. The plants were grown under an ambient relative humidity of 70–85%, with a day temperature of $22 \pm 1°C$ and night

temperature of 18 ± 1˚C, under natural light conditions, i.e., average photoperiod during cultivation was 16 hrs of daylight and 8 hrs of night. Aligned, single-stem plants measuring 25–30 cm in height with 14–18 leaves were selected for the study (Fig 1A). The collection of the VOCs and the biochemical analyses of the leaves were performed in the middle of July from plants that were at the vegetative stage of growth.

### *Rhizoctonia solani* infestation and volatile collection system

*Rhizoctonia solani* J.G. Kühn used in the experiments was isolated from an infected chrysanthemum plant and cultured on Potato Dextrose Agar (PDA) medium (Sigma Aldrich, USA) according to the common protocol. There were two types of *R. solani* inoculation applied: leaf infestation and soil inoculation. For leaf infestation, a total of 12 plants from each cultivar were sprayed with a mycelium fragments suspension that was prepared as follows: *R. solani* was grown on PDA medium in 85 mm diameter Petri plates for 14 days. Next, the mycelium suspensions were prepared by adding 20 mL of sterile water to the Petri dishes. The mycelium

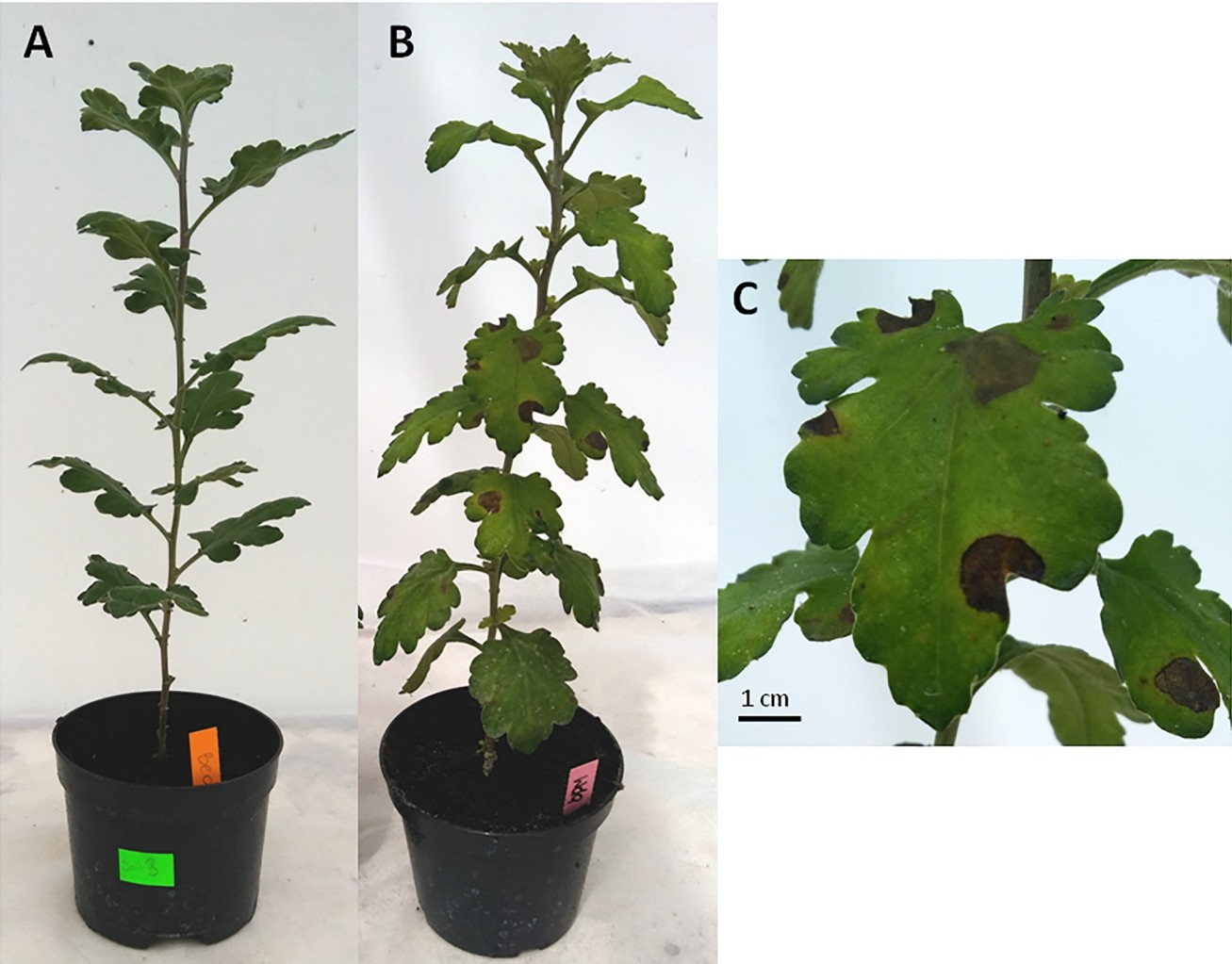

**Fig 1.** (A) The intact chrysanthemum plant prepared for infestation. (B) and (C) Showing symptoms of *Rhizoctonia solani* infestation on leaves of chrysanthemum on 6 day post leaf-inoculation.

fragments were then gently scraped from the medium surface using a spreader, washed with sterile water and filtered with a sterile gauze. Six Petri dishes, overgrown with *R. solani*, were used to prepare the suspension for 12 plants. From the total volume of 420 mL, each plant was sprayed with 35 mL of the mycelium suspension. Control plants were sprayed with distilled, sterile water without suspended *R. solani*. The VOC collection was performed on days 3 and 6 after treatment.

For soil inoculation, wheat kernels overgrown with *R. solani* were used. The preparation of kernels was as follows. Triple sterilized wheat kernels of approximately 300 mL volume were placed in sterile plastic bags. These kernels were then inoculated with five 1 $cm^2$ fragments of *R. solani* mycelium. The mycelium was cultured on the kernels for five weeks in darkness at a temperature of 22 ± 1°C. For the uniform growth of the mycelium, the kernels in closed bags were gently mixed twice a week. After the fifth week of culture, the kernels were found to be overgrown with *R. solani*. Next, the inoculated kernels were placed on the surface of the growing substrate around the plants in pots and gently mixed with the outer layer of the substrate. Six plants per cultivar were treated this way, 50 mL of *R. solani* overgrown kernels were applied to each. On day 42 post soil inoculation, the VOCs were collected.

A total of 240 chrysanthemum plants were examined for VOC emissions. From each cultivar, 12 plants were studied for leaf infestation (3 and 6 days post-infestation), six plants for in-soil infestation, and six plants were not infected by *R. solani* to serve as controls. Volatiles were collected from the entire stem of each experimental plant. Six replicates per treatment were analyzed, with a single plant serving as a replicate.

For volatile collection, the experimental chrysanthemum plants were tightly enclosed within nalophan bags, 35 cm × 60 cm, with one plant per bag. A volatile collector trap (6.35 mm outside diameter and 76 mm long glass tube (ARS, Inc., Gainesville, Florida, USA) containing 30 mg of Super-Q adsorbent (Alltech Associates, Inc., USA) provided a passive chemical filter designed for the collection of extremely low-level (ppb-ppm) VOCs from the nalophan enclosed plants. The Super-Q absorbent was inserted into each of four Tygon tubes, which were connected between the airflow meter and the collector trap. Next, purified and humidified air stream was delivered at a rate of 1.0 L $min^{-1}$ to chrysanthemum plants. To limit any contamination of volatiles from the outside of the hermetic system, a vacuum pump was used to maintain an exhaust flow of 0.8 L $min^{-1}$, resulting in a positive pressure inside the system. The collection system allowed for the simultaneous collection of volatiles from four plants, with a total collection time of 2 hrs. The potential presence of background VOCs was determined from the samples collected from empty (without plants) nalophan bags. No detectable peaks in the chromatograms (five blanks) were observed. The VOCs collection was performed in the temperature 22 ± 1°C.

The severity of the leaves' disease was determined on days 3 and 6 after leaf inoculation and day 42 after soil inoculation. The health status of the leaves was assessed on all tested plants and was expressed as the average share of leaf area showing disease symptoms (Fig 1B and 1C). The macroscopic estimation was accompanied by the analysis of fungal species identified on leaves which showed disease symptoms, which was performed after collecting the emitted volatile compounds. Isolations of the pathogens from the material on the PDA medium were made for confirmation of the identity of *R. solani* [37].

## Analytical methods

For the volatile elutions from the Super-Q adsorbent in each volatile collection trap, 225 μL of hexane was used, supplemented with 7 ng of decane as an internal standard, where the quantity of hexane used was sufficient to extract all trapped VOCs. Individual samples (1 μL) were

then injected and analyzed by GC/MS using an Auto System XL/Turbomass instrument (Perkin Elmer, USA) with a capillary column (30 m Rtx-5MS, 0.25 mm ID, 0.25 μm film thickness, Restek, USA). The temperature profile during analysis increased from 40˚C to 200˚C at a rate of 5˚C min$^{-1}$. The identification of the volatiles was verified using authentic standards (Sigma-Aldrich, USA). For the β-ocimene standard solution, both Z and E isomers were present. The emission rate (ng h$^{-1}$) of each VOC was determined by comparing the VOC's peak area relative to the peak area of the internal standard. Fourteen VOCs that were consistently detected to have a rate > 0.1 ng h$^{-1}$ are only reported in this paper, namely, (Z)-3-hexenal = (Z)-3-HAL, (E)-2-hexenal = (E)-2-HAL, (Z)-3-hexen-1-ol = (Z)-3-HOL, (E)-2-hexen-1-ol = (E)-2-HOL, (Z)-3-hexen-1-yl acetate = (Z)-3-HAC, β-pinene = β-PIN, β-myrcene = β-MYR, (Z)-ocimene = (Z)-OCI, linalool = LIN, benzyl acetate = BAC, methyl salicylate = MAT, indole = IND, β-caryophyllene = β-CAR, and (E)-β-farnesene = (E)-β-FAR.

## Biochemical array of leaves

For the extraction of chlorophylls and carotenoids, 100 mg of fresh leaf samples were taken from infected and intact plants and then homogenized in a porcelain mortar in amount of 10 mL of 100% acetone (Chemia, Poland) followed by filtration, according to the protocol elaborated by Lichtenthaler [38]. For phenolic compound evaluation, 200 mg of fresh leaf tissue were homogenized and phenolics were extracted with methanol containing 1% HCl (v/v) (Chemia, Poland), followed by the Folin–Ciocalteau [39] protocol with gallic acid (Sigma-Aldrich, USA) as the calibration standard. The analyses of the extracts were performed with the SmartSpec PlusTM spectrophotometer (BioRad, USA) at specific wavelengths of 645 nm and 662 nm for chlorophyll *a* and *b*, respectively, 470 nm for carotenoids, and 765 nm for phenolic compounds. The phenolic and pigment contents were expressed in milligrams per gram of sample fresh weight (FW). All biochemical analyses were performed three times.

## Statistical analysis

The Shapiro-Wilk's test was used to test normality of the distribution of the fourteen VOCs, i.e., for Z-3-HAL, E-2-HAL, Z-3-HOL, E-2-HOL, β-PIN, β-MYR, Z-3-HAC, (Z)-OCI, LIN, BAC, MAT, IND, β-CAR, and β-FAR and the metabolites contents in order to be able to conduct an analysis of variance (ANOVA) [40].

Since all VOCs and metabolites had normal distributions, multivariate analysis of variance (MANOVA) as well as two-way analysis of variance (ANOVA) were undertaken to determine the effects of cultivar, collection time and interaction between cultivar and collection time on VOC values [41]. Arithmetic means, standard deviations and Fisher's least significant differences (LSDs) were calculated. Pearson's linear correlation coefficients were used to assess the correlation between various VOCs at each collection time. All calculations for statistical analyses were carried out using the GenStat v.23 statistical package (VSN International, England UK).

## Results

### Results on disease rating

Clear disease symptoms in the form of necrotic spots were observed on chrysanthemum leaves inoculated using mycelium of *R. solani* (Fig 1B and 1C). The visual assessment of chrysanthemum plants with *R. solani* inoculated via the leaf, using the percentage of leaf area with disease symptoms, showed an increasing value for all cultivars during the 6 days post inoculation (dpi) recorded, with the highest value in the 'Wda' cultivar (Table 1). All chrysanthemum cultivars

**Table 1. Influence of cultivars, days past inoculation (dpi) and inoculation treatment (leaf (L) or soil (S)) with *Rhizoctonia solani* on disease symptoms incidence and re-isolation.** Fisher's least significant differences (LSDs) and values of *F*-statistics were used in the comparison of mean values for studied cultivars.

| Cultivar | Visual assessment (% of leaf area with disease symptoms) | | | *Rhizoctonia solani* re-isolation on PDA medium |
|---|---|---|---|---|
| | 3 dpi L | 6 dpi L | 42 dpi S | |
| **Ania** | 0.0±0.0 | 0.18±0.13 | 0.0±0.0 | + |
| **Beata** | 0.0±0.0 | 0.32±0.11 | 0.01±0.01 | + |
| **Brda** | 0.0±0.0 | 0.11±0.09 | 0.0±0.0 | + |
| **Kasia** | 0.0±0.0 | 0.27±0.14 | 0.02±0.01 | + |
| **Lidka** | 0.0±0.0 | 0.36±0.17 | 0.0±0.0 | + |
| **Luczniczka** | 0.0±0.0 | 0.15±0.08 | 0.0±0.0 | + |
| **Malgosia** | 0.0±0.0 | 0.21±0.10 | 0.0±0.0 | + |
| **Polka** | 0.14±0.13 | 1.24±0.46 | 0.03±0.02 | + |
| **Wda** | 0.76±0.35 | 4.78±1.66 | 0.22±0.12 | + |
| **Zofia** | 0.0±0.0 | 0.03±0.02 | 0.0±0.0 | + |
| LSD$_{0.05}$ | 0.137 | 0.643 | 0.045 | |
| *F*-statistic | 19.21*** | 41.05*** | 18.60*** | |

exhibited disease symptoms at 6 dpi, and only 2 cultivars at 3 dpi. Soil inoculation of the chrysanthemum plants resulted in 4 cultivars showing disease symptoms on leaves at 42 dpi; however, the percentage of the leaf area with disease symptoms was lower compared to leaf inoculation. Re-isolation of samples from the diseased plants on PDA were successful in all chrysanthemum plants inoculated with *R. solani*.

## VOCs

The analysis of the background (control) VOC emissions indicated differences in both the quality and the quantity of the produced odor between tested chrysanthemum cultivars (Table 2). The bouquet of all fourteen VOCs investigated were produced by only four cultivars, namely, 'Ania', 'Kasia', 'Luczniczka' and 'Wda'. Seven out of fourteen detected volatiles were present in all tested cultivars, namely: Z-3-HAL, E-2-HAL, Z-3-HAC, (Z)-OCI, LIN, β-CAR, and β-FAR. Conversely, five out of the fourteen VOCs were found to be not emitted by the 'Brda' and 'Zofia' cultivars. For other cultivars, several of the fourteen identified VOCs were not detected ('Lidka'– 4 VOCs, 'Beata'– 3 VOCs, 'Polka' and 'Malgosia'– 2 VOCs) (Table 2). The least representative were β-PIN and β-MYR, which were absent in four cultivars ('Beata', 'Lidka', 'Polka' and 'Zofia') and MAT and IND that were also absent in four cultivars ('Beata', 'Lidka', 'Malgosia' and 'Zofia'). The absence of particular VOCs being emitted by the control plants was confirmed after infestations. They were also not produced by infected plants.

Considerable differences were observed between cultivars in terms of their reaction to *R. solani* infection. 'Beata' was the cultivar emitting the highest amounts of VOCs, whereas cultivar 'Kasia' produced the least amount of VOCs, which meant its reaction to *Rhizoctonia* was the smallest (Table 2). Analysis of variance indicates that the main effects of cultivars were significant for all of the VOCs for control data, except E-2-HOL and BAC (Table 2). Cultivar (Wilk's λ = 0.0008; $F_{42;140}$ = 33.39), collection time after infestation (Wilk's λ = 0.0019; $F_{28;94}$ = 73.48), as well as cultivar × collection time after infestation interaction (Wilk's λ = 0.00016; $F_{84;268}$ = 12.21) were significant ($p < 0.0001$) for all fourteen VOC compounds jointly. ANOVA indicated that the main effects of cultivar, collection time after the infestation and cultivar × collection time interaction were significant for all the VOCs of study (S1 Table).

**Table 2. Mean values of the VOC emission rates (ng hr$^{-1}$) and standard deviations for ten cultivars for all of the fourteen VOCs.** Fisher's least significant differences (LSDs) and values of $F$-statistics were used in the comparison of mean values for studied cultivars (*means of cultivars; statistically significant difference).

| Trait | Ania | Beata | 'Brda' | Kasia | Lidka | Luczniczka | Malgosia | Polka | Wda | Zofia | LSD$_{0.05}$ | $F$-statistic |
|---|---|---|---|---|---|---|---|---|---|---|---|---|
| Z-3-HAL | 4.9±1.3 | 30.6±7.5 | 3.7±1.2 | 3.2±1.3 | 7.4±1.7 | 12.5±2.4 | 6.7±5.2 | 7.6±2.2 | 11.9±2.0 | 4.3±3.9 | 4.02 | 33.28*** |
| E-2-HAL | 2.9±0.8 | 9.6±3.9 | 4.1±2.4 | 1.8±1.0 | 4.2±1.1 | 5.3±1.7 | 4.8±5.0 | 5.3±2.0 | 5.9±2.3 | 1.8±0.9 | 2.88 | 5.09*** |
| Z-3-HOL | 1.2±0.8 |  | 3.8±1.8 | 1.7±1.3 | 3.5±2.0 | 2.7±1.0 | 4.4±2.9 | 4.5±1.5 | 4.3±2.3 | 2.8±0.3 | 1.99 | 2.87* |
| E-2-HOL | 3.5±2.8 |  | 2.7±0.7 | 2.4±0.5 | 2.8±0.9 | 3.3±0.6 | 5.4±4.0 | 2.4±0.8 | 3.2±1.9 | 2.8±1.1 | 2.16 | 1.47 |
| β-PIN | 3.3±1.1 | 8.1±3.9 |  | 2.7±0.5 |  | 4.8±2.2 | 5.2±3.7 |  | 5.0±1.8 |  | 2.97 | 3.37* |
| β-MYR | 3.9±1.8 | 5.1±1.6 |  | 0.9±0.5 |  | 2.5±0.5 | 4.8±1.9 |  | 1.9±0.8 |  | 1.56 | 9.74*** |
| Z-3-HAC | 4.4±1.0 | 12.8±7.4 | 2.6±2.1 | 3.5±3.2 | 5.2±2.4 | 6.7±1.1 | 4.8±2.4 | 9.6±2.3 | 15.2±6.6 | 2.0±0.7 | 4.22 | 8.97*** |
| (Z)-OCI | 2.4±0.6 | 18.8±4.9 | 3.2±0.9 | 1.0±0.6 | 6.3±2.7 | 6.3±0.9 | 4.7±1.1 | 6.2±1.2 | 12.4±4.2 | 3.2±1.6 | 2.77 | 30.02*** |
| LIN | 3.2±0.3 | 15.0±4.0 | 2.2±0.5 | 1.5±1.0 | 5.3±2.0 | 6.8±1.7 | 4.1±1.1 | 4.4±1.3 | 9.7±5.5 | 2.8±1.7 | 2.86 | 16.76*** |
| BAC | 2.3±0.8 |  |  | 2.8±0.6 | 3.6±3.3 | 3.5±0.5 | 5.4±3.8 | 3.0±1.1 | 4.4±1.5 |  | 2.43 | 1.57 |
| MAT | 2.1±0.6 | 4.5±1.6 |  | 3.2±0.7 |  | 4.3±1.8 |  | 2.6±1.1 | 3.7±0.6 |  | 1.37 | 3.86** |
| IND | 2.8±0.7 | 6.6±1.2 |  | 1.9±0.9 |  | 2.0±2.0 |  | 1.7±0.8 | 1.7±0.9 |  | 1.37 | 16.46*** |
| β-CAR | 3.6±2.7 | 10.0±4.0 | 1.6±0.7 | 1.7±0.7 | 4.7±2.6 | 4.8±1.0 | 4.0±1.7 | 3.9±1.4 | 4.6±2.2 | 2.7±0.6 | 2.38 | 7.84*** |
| β-FAR | 1.7±1.0 | 11.1±4.7 | 2.7±0.8 | 3.2±1.1 | 2.9±1.1 | 5.6±1.6 | 3.6±1.7 | 4.5±1.9 | 4.7±1.4 | 1.9±1.2 | 2.29 | 11.5*** |

Mean values of the emission rates of the observed VOCs for the studied cultivars are presented in supporting information (S1–S14 Figs). Provided the analyzed VOC was produced by the cultivar, the highest emission rate was mostly found on day 3 post leaf-infection with *R. solani*, and the lowest on day 42 after soil administration of infected wheat kernels. Two exceptions to this rule were observed. The 'Kasia' cultivar produced four times higher emission rates of E-2-HOL on day 6 after infestation rather than on day 3 post treatment. Similarly, β-PIN emission rate in 'Beata' was higher on day 6 after infestation than on day 3 (21.3 and 15.2 ng hr$^{-1}$, respectively).

Results of the correlation analyses between VOC emissions assessed on days 3 and 6 post inoculation, as well as on day 42 following soil inoculation, independently, are depicted as heatmaps in Figs 2–4, respectively, with dark green depicting the highest and red the lowest mean emission rates.

Significant positive relationships in all three time periods, i.e., on days 3 and 6 post leaf inoculation and on day 42 post soil inoculation, were observed between the following pairs of VOCs: Z-3-HAL and Z-3-HAC, Z-3-HAL and (Z)-OCI, Z-3-HAL and β-FAR, Z-3-HAC and (Z)-OCI, Z-3-HOL and MAT, Z-3-HAC and β-FAR (Figs 2–4). Different signs of correlation coefficients were observed between E-2-HAL and MAT (negative on day 3 post inoculation, positive on day 6 post inoculation and on day 42 post soil inoculation) (Figs 2–4). Significant positive relationships on days 3 and 6 post inoculation were observed for Z-3-HOL and β-MYR, β-CAR and β-FAR, (Z)-OCI and LIN, Z-3-HOL and BAC as well as β-MYR and β-FAR (Figs 2 and 3). Different signs of correlation coefficients were found for LIN and E-2-HOL, LIN and β-MYR, as well as LIN and β-FAR (positive on day 3 and negative on day 6 post inoculation) (Figs 2 and 3). Z-3-HAL and E-2-HOL, E-2-HOL and Z-3-HAC, β-MYR and MAT, LIN and β-CAR were positively correlated on day 3 post inoculation and on day 42 post soil inoculation (Figs 2 and 4). On day 6 post inoculation and on day 42 post soil inoculation positive correlations between E-2-HOL and β-FAR are found, whereas negative correlations between Z-3-HAL and MAT, Z-3-HOL and (Z)-OCI, (Z)-OCI and MAT occur (Figs 3 and 4). Some correlations are significant only in one time period: (1) on day 3 post inoculation—E-2-HAL and β-PIN, E-2-HOL and β-PIN, Z-3-HAL and LIN, Z-3-HAL and β-CAR, E-2-HOL and (Z)-OCI, BAC and MAT, E-2-HOL and β-CAR, Z-3-HAC and LIN, Z-3-HAC and β-

CAR, β-MYR and BAC, (Z)-OCI and β-CAR, (Z)-OCI and β-FAR (positive), E-2-HOL and MAT, β-PIN and BAC, β-PIN and MAT (negative) (Fig 2); (2) on day 6 post inoculation—Z-3-HAL and β-PIN, E-2-HOL and β-MYR, β-PIN and (Z)-OCI, MAT and IND, MAT and β-CAR, E-2-HAL and IND, β-PIN and LIN, β-MYR and β-CAR (positive), Z-3-HOL and β-PIN, β-PIN and β-MYR, Z-3-HOL and LIN, (Z)-OCI and BAC, LIN and BAC (negative) (Fig 3); (3)

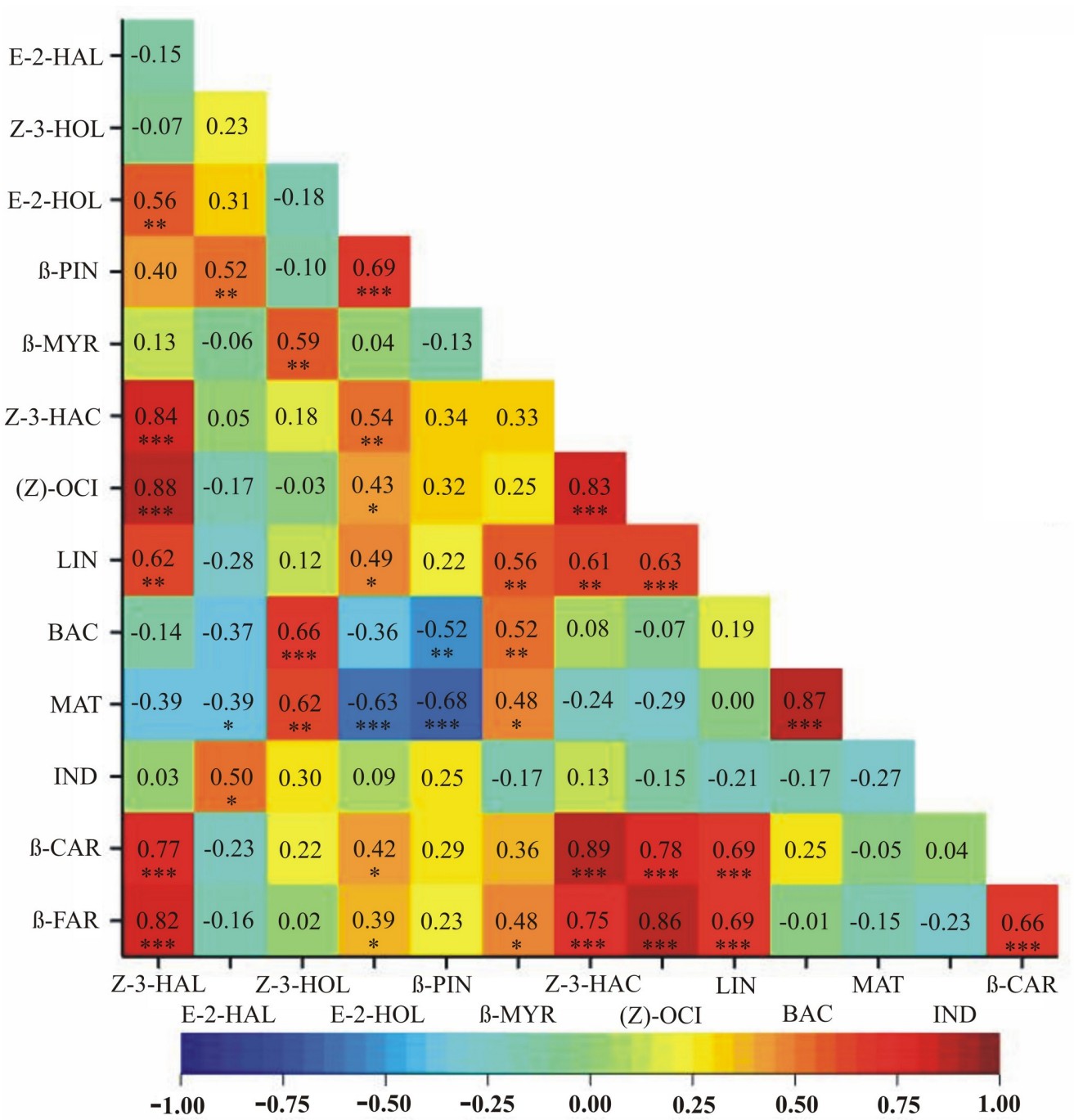

**Fig 2. A heatmap showing correlation coefficients between VOC emissions assessed on day 3 post leaf-inoculation.** Dark green depicts the highest and red the lowest mean emission rates. * $p < 0.05$; ** $p < 0.01$; *** $p < 0.001$.

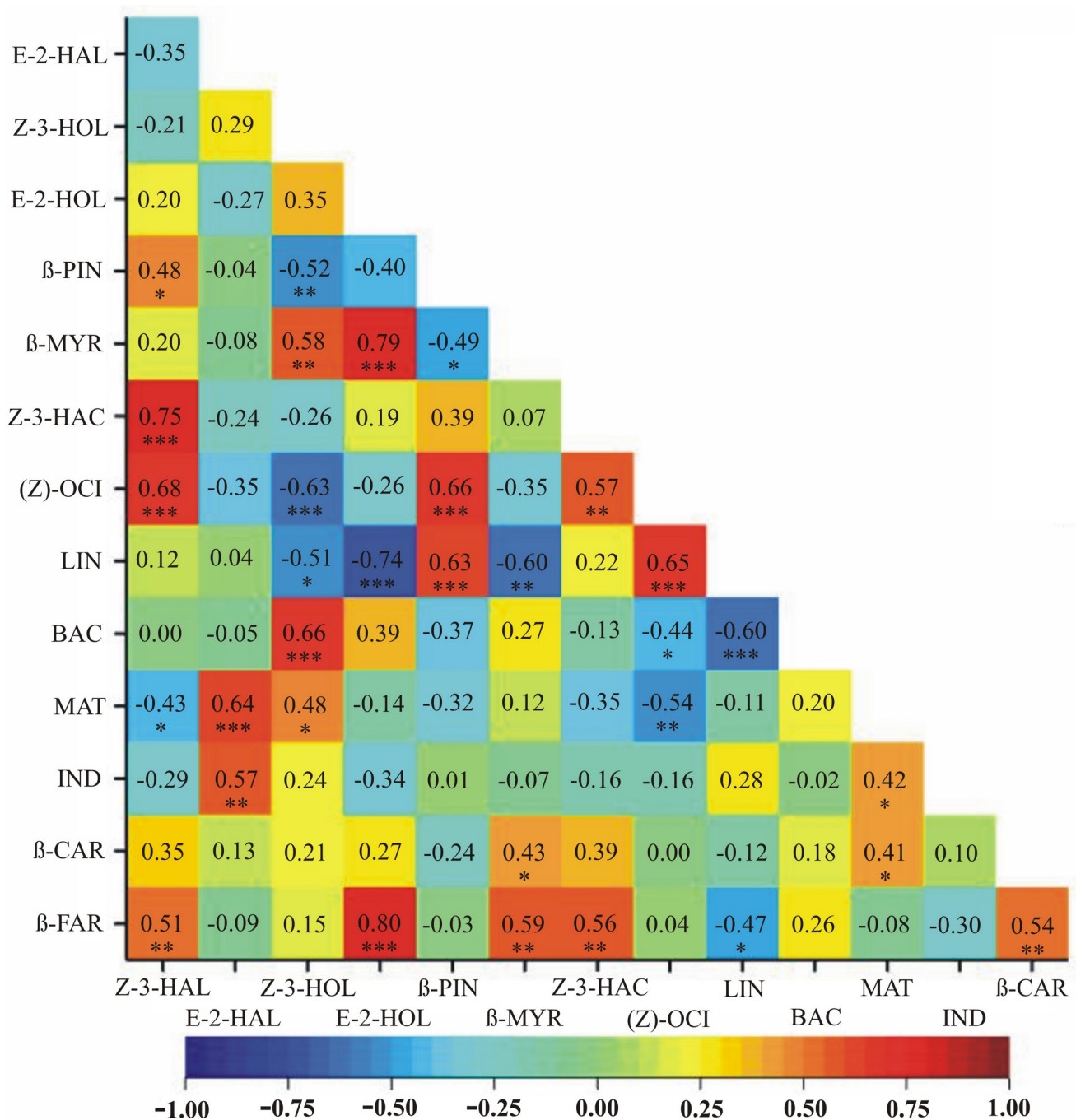

**Fig 3. A heatmap showing correlation coefficients between VOC emissions assessed on day 6 post leaf-inoculation.** Dark green depicts the highest and red the lowest mean emission rates. * $p < 0.05$; ** $p < 0.01$; *** $p < 0.001$.

on day 42 post soil inoculation—E-2-HAL and Z-3-HOL, IND and β-FAR, E-2-HAL and β-CAR, β-PIN and β-CAR, LIN and MAT (positive), Z-3-HAL and Z-3-HOL, Z-3-HOL and E-2-HOL, Z-3-HAC and E-2-HAL, Z-3-HAC and Z-3-HOL, E-2-HAL and (Z)-OCI, Z-3-HAC and MAT (negative) (Fig 4).

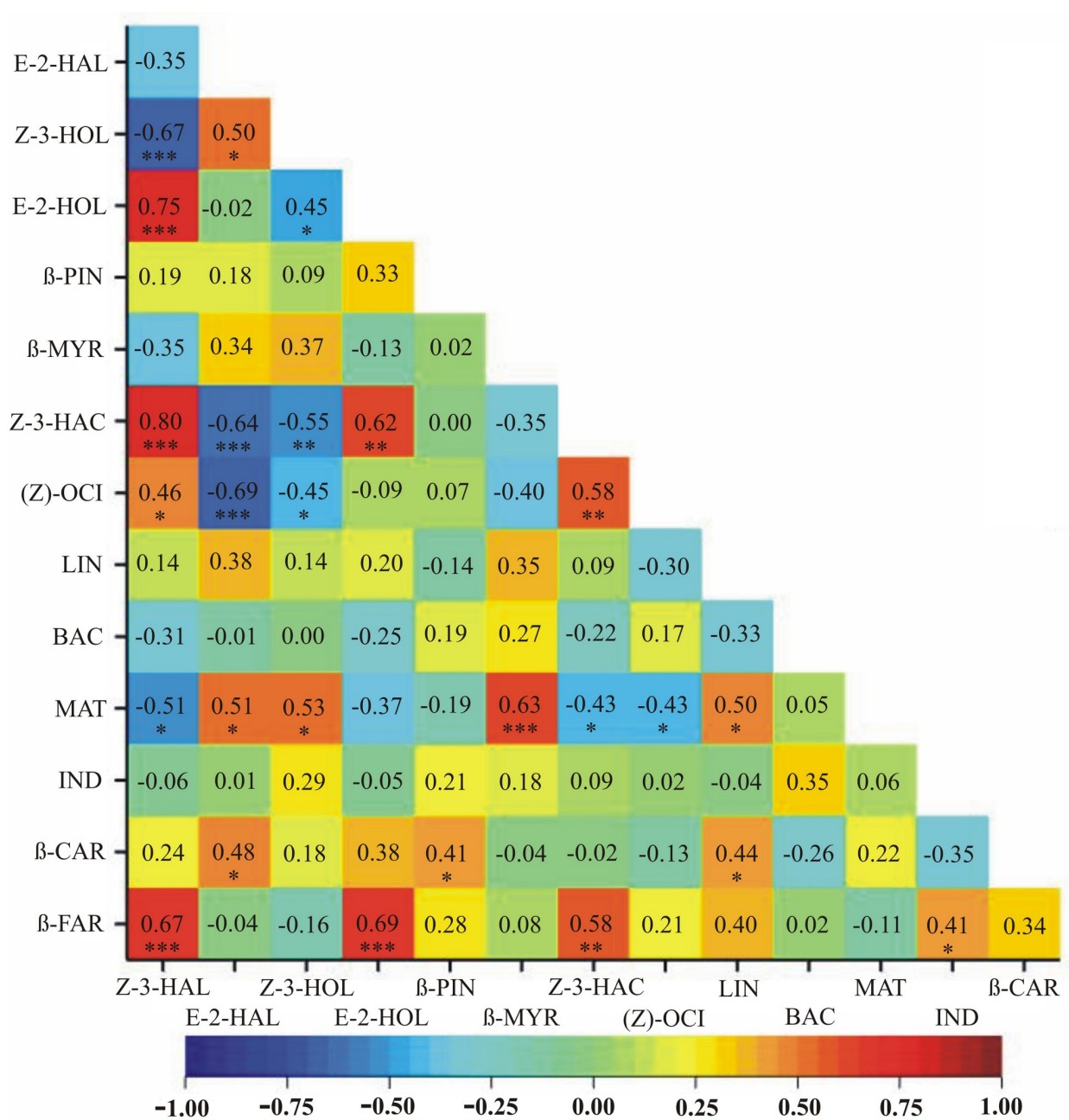

**Fig 4. A heatmap showing correlation coefficients between VOC emissions assessed on day 42 after soil inoculation.** Dark green depicts the highest and red the lowest mean emission rates. * $p < 0.05$; ** $p < 0.01$; *** $p < 0.001$.

## Analysis of the chlorophylls, carotenoids and phenolic compounds content

As for the results of leaves' biochemical composition, the two-way ANOVA analysis revealed significant differences between the tested cultivars and *R. solani* treatments (Tables 3 and S3).

The mean chlorophyll *a* content in different cultivars, regardless the *R. solani* treatments, ranged between 0.73 mg g$^{-1}$ FW in 'Zofia' and 1.09 mg g$^{-1}$ FW in 'Brda'. The lowest (0.30 mg g$^{-1}$ FW) and the highest (0.47 mg g$^{-1}$ FW) mean content of chlorophyll *b* was also found in

'Zofia' and 'Brda' cultivars, respectively. The soil infestation with *R. solani* resulted in a significant decrease in the mean chlorophyll *a* content as compared to the control, whereas the leaves' infestation enhanced the biosynthesis of chlorophyll *b*. Nevertheless, the *R. solani* treatments did not affect the total content of chlorophylls (*a+b*). However, this trait varied significantly between the tested cultivars (Table 3). Significant differences were also observed between the tested cultivars in terms of the chlorophyll *a*/*b* ratio, as well as the chlorophylls (*a+b*)/carotenoids' ratio. The leaves and soil infestations with *R. solani* decreased the chlorophyll *a*/*b* ratio, whereas the chlorophyll (*a+b*)/carotenoids ratio was significantly higher for leaf infestation as compared to control and soil infestation (S2 and S3 Tables).

*R. solani* infestations decreased the content of carotenoids. The most intensive accumulation of these metabolites, irrespective of the *R. solani* treatment, was found in 'Brda' (0.26 mg g$^{-1}$ FW) and 'Lidka'/'Polka' (0.24 mg g$^{-1}$ FW) leaves. Cultivar 'Zofia' (0.17 mg g$^{-1}$ FW) was characterized by the lowest mean content of carotenoids in its leaves (Table 3).

The most intensive biosynthesis of phenolics was found in leaves of plants growing in an infested substrate (12.73 mg g$^{-1}$ FW), while in control and leaf-infested plants the mean contents of phenolic compound were similar (7.83 and 8.11 mg g$^{-1}$, respectively). The mean content of total phenolics was found to be higher in 'Kasia' (13.38 mg g$^{-1}$ FW) than in 'Ania', 'Beata', 'Lidka', 'Wda', 'Zofia' and 'Luczniczka' (Table 3).

## Discussion

*Rhizoctonia* species are soil-borne fungal pathogens causing root and leaf diseases on a wide spectrum of crop species including ornamentals [42]. Therefore, *Rhizoctonia* infection poses a significant challenge in agriculture, prompting efforts to address this issue. Brown necroses on leaves, root and stem rot and damping-off cuttings are typical symptoms of *Rhizoctonia* infection on chrysanthemum, which leads to a decrease of the plants quality, thereby resulting in production losses [43]. Since fungicide applications are not environmentally- or health-friendly, novel and safer methods for pathogen control are in demand. Physical methods such as soil solarization or electron-beam treatment have been shown to be efficient in *Rhizoctonia* management in chrysanthemum production [44,45]. Other methods focus on an application of natural antagonists of the fungus called plant growth promoting rhizobacteria with or without biostimulants, e.g., seaweed extracts to promote a plant's natural defense mechanism [46]. Other novel approaches involve a plant's inner ability to control the invader by intrinsic, genetically based mechanisms, to which the emission of specific compounds in response to pathogen infestation belongs [37].

Particular cultivars of chrysanthemum in our experiment varied in terms of both the type and quantity of VOCs produced as a result of *R. solani* infestation. In many plants, pathogen infection induces the synthesis of volatile terpenes. For example, a higher emission of terpenes was observed in susceptible poplar cultivars infected by the rust fungus *Melampsora laricipopulina* compared to healthy plants [47,48]. On the other hand, the positive correlations between VOC emissions by plants and pathogens resistance have been revealed in several other studies. For example, grapevine genotypes resistant to downy mildew (*Plasmopara viticola*) produced significantly higher quantities of monoterpenes and sesquiterpenes compared to susceptible genotypes [49].

Rice genotypes resistant to the bacterial pathogen *Xanthomonas oryzae* pv. *oryzae* emitted large quantities of either the sesquiterpene (E)-nerolidol [50] or the monoterpene (S)-limonene [51]. In citrus, the tolerance to huanglongbing disease was also associated with higher C6 aldehydes (GLVs) and monoterpene emissions [52]. Similarly, sesquiterpene (E)-b-caryophyllene, which is the major VOC emitted from *Arabidopsis thaliana* flowers, is a defense against a

**Table 3. Primary (chlorophyll *a* and chlorophyll *b*) and secondary (carotenoid and phenolic compounds) metabolites content (mg g$^{-1}$ FW) in leaves of ten chrysanthemum cultivars infested with *Rhizoctonia solani* on leaves and on the surface of growth substrate.** Mean values and standard deviations (s.d.) in columns and rows followed by the same letter do not differ significantly with two-way ANOVA and Fisher's post-hoc test at $p < 0.05$. Capital letters refer to the main effects (irrespectively), small letters refer to the interaction between the two studied independent variables.

| *Rhizoctonia solani* treatment | | Ania | Beata | Brda | Kasia | Lidka | Luczniczka | Malgosia | Polka | Wda | Zofia | Mean |
|---|---|---|---|---|---|---|---|---|---|---|---|---|
| | | | | | | | Cultivars | | | | | |
| | | | | | | | Chlorophyll *a* | | | | | |
| control | Mean | 0.87 d-h | 0.82 e-i | 1.16 a | 0.93 c-f | 0.99 a-e | 0.93 c-f | 0.90 c-h | 1.08 a-c | 0.94 c-f | 0.85 d-h | **0.95 A** |
| | s.d. | 0.01 | 0.14 | 0.10 | 0.08 | 0.09 | 0.09 | 0.11 | 0.16 | 0.08 | 0.11 | |
| leaves infestation | Mean | 0.72 g-i | 0.92 c-f | 1.15 a,b | 0.92 c-f | 1.02 a-d | 0.98 a-e | 0.84 d-h | 0.93 c-f | 0.85 d-h | 0.70 h-i | **0.90 AB** |
| | s.d. | 0.15 | 0.05 | 0.10 | 0.08 | 0.07 | 0.10 | 0.13 | 0.07 | 0.07 | 0.11 | |
| soil infestation | Mean | 0.84 d-h | 0.99 a-e | 0.96 b-e | 0.92 c-f | 0.92 c-f | 1.00 a-e | 0.97 a-e | 0.80 e-i | 0.76 f-i | 0.64 i | **0.88 B** |
| | s.d. | 0.16 | 0.10 | 0.09 | 0.19 | 0.20 | 0.03 | 0.14 | 0.06 | 0.26 | 0.09 | |
| **Mean** | | **0.81 DE** | **0.91 B-D** | **1.09 A** | **0.92 BC** | **0.98 AB** | **0.97 B** | **0.90 B-D** | **0.94 BC** | **0.85 CD** | **0.73 E** | |
| | | | | | | | Chlorophyll *b* | | | | | |
| control | Mean | 0.39 b-f | 0.26 f | 0.47 a-d | 0.48 a-c | 0.33 c-f | 0.29 e,f | 0.36 b-f | 0.37 b-f | 0.36 b-f | 0.29 e,f | **0.36 B** |
| | s.d. | 0.14 | 0.09 | 0.03 | 0.20 | 0.05 | 0.08 | 0.13 | 0.12 | 0.08 | 0.08 | |
| leaves infestation | Mean | 0.32 d-f | 0.43 a-e | 0.56 a | 0.44 a-e | 0.50 a,b | 0.43 a-e | 0.38 b-f | 0.41 a-e | 0.38 b-f | 0.32 d-f | **0.42 A** |
| | s.d. | 0.10 | 0.05 | 0.04 | 0.11 | 0.07 | 0.07 | 0.05 | 0.04 | 0.04 | 0.06 | |
| soil infestation | Mean | 0.37 b-f | 0.43 a-e | 0.39 b-f | 0.38 b-f | 0.37 b-f | 0.42 a-e | 0.38 b-f | 0.33 d-f | 0.31 e,f | 0.30 e,f | **0.37 B** |
| | s.d. | 0.16 | 0.06 | 0.12 | 0.08 | 0.12 | 0.07 | 0.08 | 0.02 | 0.14 | 0.04 | |
| **Mean** | | **0.36 BC** | **0.37 BC** | **0.47 A** | **0.43 AB** | **0.40 AB** | **0.38 BC** | **0.37 BC** | **0.37 BC** | **0.35 BC** | **0.30 C** | |
| | | | | | | | Chlorophylls (*a* + *b*) | | | | | |
| control | Mean | 1.26 c-g | 1.07 e-g | 1.63 ab | 1.41 a-d | 1.32 b-f | 1.22 c-g | 1.25 c-g | 1.45 a-d | 1.30 b-f | 1.14 d-g | **1.31 A** |
| | s.d. | 0.15 | 0.22 | 0.13 | 0.28 | 1.14 | 0.17 | 0.24 | 0.28 | 0.14 | 0.19 | |
| leaves infestation | Mean | 1.04 e-g | 1.35 b-f | 1.71 a | 1.36 b-e | 1.53 a-c | 1.41 a-d | 1.23 c-g | 1.35 b-f | 1.23 c-g | 1.03 f,g | **1.32 A** |
| | s.d. | 0.26 | 0.09 | 0.12 | 0.18 | 0.10 | 0.17 | 0.18 | 0.10 | 0.11 | 0.16 | |
| soil infestation | Mean | 1.20 c-g | 1.42 a-d | 1.35 b-f | 1.30 b-f | 1.30 b-f | 1.42 a-d | 1.35 b-f | 1.13 d-g | 1.07 e-g | 0.94 g | **1.25 A** |
| | s.d. | 0.31 | 0.14 | 0.19 | 0.27 | 0.31 | 0.08 | 0.22 | 0.05 | 0.40 | 0.12 | |
| **Mean** | | **1.17 CD** | **1.28 BC** | **1.56 A** | **1.36 B** | **1.38 AB** | **1.35 BC** | **1.28 BC** | **1.31 BC** | **1.20 B-D** | **1.04 D** | |
| | | | | | | | Carotenoids | | | | | |
| control | Mean | 0.18 g-l | 0.21 d-i | 0.27 a | 0.22 c-i | 0.25 a-d | 0.23 a-f | 0.23 a-f | 0.27 a | 0.22 c-i | 0.21 d-i | **0.23 A** |
| | s.d. | 0.02 | 0.03 | 0.02 | 0.01 | 0.02 | 0.02 | 0.01 | 0.02 | 0.02 | 0.03 | |
| leaves infestation | Mean | 0.17 j-l | 0.23 a-f | 0.26 a-c | 0.21 d-i | 0.23 a-f | 0.22 c-i | 0.21 d-i | 0.22 c-i | 0.18 g-l | 0.16 k,l | **0.21 B** |
| | s.d. | 0.04 | 0.01 | 0.04 | 0.02 | 0.01 | 0.01 | 0.05 | 0.02 | 0.02 | 0.02 | |
| soil infestation | Mean | 0.19 f-k | 0.23 a-f | 0.23 a-f | 0.23 a-f | 0.23 a-f | 0.23 a-f | 0.24 a-e | 0.20 e-i | 0.18 g-l | 0.14 l | **0.21 B** |
| | s.d. | 0.03 | 0.01 | 0.02 | 0.03 | 0.05 | 0.01 | 0.04 | 0.02 | 0.06 | 0.03 | |
| **Mean** | | **0.18 D** | **0.23 B** | **0.26 A** | **0.22 BC** | **0.24 AB** | **0.23 B** | **0.23 B** | **0.24 AB** | **0.19 CD** | **0.17 D** | |
| | | | | | | | Phenolic compounds | | | | | |
| control | Mean | 6.85 c-e | 8.48 b-e | 10.52 b-e | 10.20 b-e | 7.99 b-e | 6.42 c-e | 7.63 b-e | 8.31 b-e | 5.52 d,e | 6.34 c-e | **7.83 B** |
| | s.d. | 1.16 | 1.95 | 3.56 | 1.49 | 1.97 | 0.34 | 0.60 | 2.14 | 0.89 | 2.36 | |
| leaves infestation | Mean | 7.43 b-e | 8.81 b-e | 9.21 b-e | 6.94 c-e | 6.87 c-e | 6.60 c-e | 7.36 b-e | 10.88 b-e | 6.86 c-e | 10.17 b-e | **8.11 B** |
| | s.d. | 1.19 | 0.92 | 0.55 | 1.66 | 0.98 | 2.26 | 1.29 | 2.50 | 2.33 | 1.04 | |
| soil infestation | Mean | 13.63 b | 9.93 b-e | 11.99 b-d | 23.00 a | 11.94 b-d | 5.92 c-e | 23.73 a | 12.07 b,c | 10.85 b-e | 4.98 e | **12.73 A** |
| | s.d. | 1.48 | 1.09 | 3.60 | 9.09 | 1.89 | 0.34 | 4.21 | 3.50 | 0.39 | 1.12 | |
| **Mean** | | **9.30 B-D** | **9.07 CD** | **10.57 A-C** | **13.38 A** | **8.93 CD** | **6.31 D** | **12.91 AB** | **10.42 A-C** | **7.74 CD** | **7.16 CD** | |

bacterial pathogen [53]. In our study, only four cultivars, namely 'Ania', 'Kasia', 'Luczniczka' and 'Wda' emitted all of the fourteen VOCs, which, based on the experiments cited above, may suggest that the four cultivars, emit a wide spectrum of VOCs to prevent pathogen infection.

Interestingly, the emission of green leaf volatiles (GLVs) was proven to adversely affect necrotrophic fungal pathogens. This may result from induction of jasmonate-mediated

signaling cascades, which also induce the emission of GLVs and additionally trigger effective defense responses against these pathogens. Similar to terpenoids, several studies have revealed positive associations between GLV emission and resistance to pathogens. For instance, in maize kernels, a positive correlation was found between resistance to *Aspergillus flavus* infection and the presence of (Z)-hexenal and (Z)-decenal [54]. Also, it was reported that when spores of *Colletotrichum lindemuthianum* were exposed to the volatiles emitted by a resistant bean genotype that produced high levels of nonanal and other volatiles, spore germination was irreversibly inhibited [29]. Moreover, *in vivo* evidence obtained through functional genetic approaches demonstrated that these compounds can directly exhibit toxicity towards pathogens during the infection process. For example, transgenic tomato or *Arabidopsis* plants that overproduced GLVs exhibited significantly higher resistance to *Alternaria alternata* f. sp. *lycopersici* [55] or *Botrytis cinerea* [56], respectively, compared to wild-type plants.

It is worth noticing that seven out of the fourteen volatiles were present in all tested cultivars, namely: Z-3HAL. E-2-HAL, Z-3-HAC, (Z)-OCI, LIN, β-CAR, and β-FAR. These VOCs are involved in various functions in plants. Z-3-HAL is an aroma compound commonly found in fruits such as apples and pears. It contributes to the characteristic green or grassy scent of these fruits [57]. E-2-HAL is a volatile compound responsible for the characteristic aroma of fresh-cut grass. It also acts as a signaling molecule in plants, playing a role in stress responses and defense against pathogens [58]. Z-3-HAC is an ester that contributes to the aroma of fresh, green vegetation. It is often found in fruits, vegetables, and herbs and is associated with a pleasant, fruity scent [59] (Z)-OCI is a volatile terpene that contributes to the aroma of various flowers, including orchids, lavender, and roses. It has a sweet, floral scent and can attract pollinators [60]. LIN is a naturally occurring terpene alcohol found in many flowers and spice plants [61]. It has a sweet, floral aroma and is commonly used in perfumes and aromatherapy. LIN may also possess anti-inflammatory and sedative properties [62]. β-CAR is a natural sesquiterpene compound found in various plants, particularly in essential oils derived from spices such as black pepper, cloves, and oregano. It exhibits a range of biological functions and has been studied for its potential therapeutic and antimicrobial properties [63]. And finally, β-FAR is a sesquiterpene commonly found in apples and other fruits. It contributes to the fruity aroma and may have insecticidal properties, acting as a natural repellent against certain pests [64]. Whilst all seven VOCs contribute to fresh and fruity aromas of plants, only four of them E-2-HAL, LIN, β-CAR, and β-FAR have the potential to be used by plants for protection against pathogens, because they contribute to antioxidant and anti-inflammatory properties. Nevertheless, all of those seven VOCs were constantly detected in ten of the chrysanthemum cultivars investigated. Conversely, four out of the fourteen VOCs, namely β-PIN, β-MYR, IND, and MAT, were not detected to be emitted by the 'Brda', 'Lidka' and 'Zofia' cultivars (Table 2). This is interesting since β-PIN and β-MYR are connected to antimicrobial properties [65,66], and IND is a molecule involved in a plant's defense responses [61,62,67]. Moreover, the β-PIN and β-MYR were found to be the least representative, being absent in four cultivars, 'Beata', 'Lidka', 'Polka', and 'Zofia'. Similarly MAT and IND were found to be absent in four cultivars, 'Beata', 'Lidka', 'Malgosia' and 'Zofia'. Since these VOCs were not produced by infected plants, it suggests that their role in chrysanthemum's defense is menial.

Considerable differences were recorded between the *Chrysanthemum* cultivars tested in terms of their reaction to *R. solani* infection. The highest amount of VOCs was emitted by 'Beata', whereas cultivar 'Kasia' produced the least amount of VOCs, which means that its reaction to *R. solani* was the smallest (Table 2). Correlation analyses showed variations in VOC emissions at different time points after infection with *R. solani*, with some exceptions observed in 'Kasia' and 'Beata' cultivars. The variation in the amount of VOCs released by plants during fungal pathogen infection can depend on several factors, including the specific plant species,

the type of fungal pathogen, the stage of infection, environmental conditions, and the composition of the neighbouring plant community [68]. The actual amount of VOCs released by plants during fungal pathogen infection is challenging to quantify precisely due to the complex nature of the interactions and the difficulty in measuring VOCs in real-time [69]. Moreover, the specific composition and quantity of VOCs emitted can vary between plant species, pathogen strains, and stages of infection [68]. This study has shown correlations between VOCs emissions assessed at days 3 and 6 post inoculation, as well as on day 42 post soil inoculation, independently (Figs 2–4). It was found that on day 3 post inoculation the emission of Z-3-HAL was correlated with the emissions of Z-3-HAC, (Z)-OCI, β-CAR and β-FAR (Fig 2). All of the abovementioned VOCs are aroma compounds and are responsible for the pleasant grassy and fruity scent of chrysanthemum [57,59,60,63,64]. The only exception is β-CAR, also known as caryophyllene, which is a natural sesquiterpene compound found in numerous important aromatic plants. It exhibits a range of biological functions and has been studied for its potential therapeutic properties and is described as a dietary cannabinoid [70] with antibacterial properties [53]. Similarly, on day 6 post inoculation, the correlated emission of the following VOCs was found: Z-3-HAL, Z-3-HAC, (Z)-OCI and E-2-HOL together with β-FAR (Fig 3). The correlation of aroma compounds such as Z-3-HAL, Z-3-HAC and (Z)-OCI, was maintained. The significant correlation of aroma compounds facilitates the selection of materials. In contrast, the lack of correlation makes the process very complicated. Especially when more than a dozen compounds are considered simultaneously. The results obtained indicate the advisability of conducting studies involving the analysis of multiple VOCs simultaneously. Additionally, the emissions of E-2-HOL and β-FAR were increased. Both those VOCs are associated not only with fresh aroma but also with response to herbivore damage or other stress (E-2-HOL) [58] and with insecticidal properties, being a natural repellent against certain pests (β-FAR) [63,64]. On day 42 post soil inoculation, the highest correlation was recorded for the emissions of Z-3-HAL, E-2-HOL and Z-3-HAC (Fig 4). All three VOCs contribute to the observed fresh and fruity aroma [57,59,71]. Among them, only E-2-HOL is released by the plant in response to herbivore attack [71]. These results are congruent with another study where it was recorded that chrysanthemum plants infected with the fungus *Botrytis cinerea* reacted by inducing its defense system and stimulating the emission of VOCs involved in deterring herbivore attack [18]. Other studies also report a disease-specific emission of VOCs as a response of plants to pathogen attack, e.g., in the case of apple tree infection with *Erwinia amylovora* bacterium [72]. The specific VOCs released could be also a way for the chrysanthemum plants to deter insect herbivores or attract natural enemies on the herbivore. In olfactometer bioassay experiments, aphids showed a preference for the odor of both healthy and infested *Chrysanthemum morifolium*. However, the opposite result was recorded for *Artemisia annua*. Nevertheless, with time the aphids were found to be attracted more to the healthy plants than to the infested plants [73].

The interactions of VOCs present in the environment are complex, and it should be noted that the soil is a huge reservoir and source of biogenic VOCs [74], which makes it very difficult to predict all consequences of various VOCs interactions. Nevertheless, VOCs emitted by plants are studied extensively for their potential use in pest biocontrol and disease management [75], and for applications to develop sustainable defense strategies and productivity of crops in agriculture [76].

In this study, we have examined how *R. solani* infestation affects the emission of VOCs from chrysanthemum plants. As part of our investigation, we measured also the concentrations of chlorophyll, carotenoids, and total phenolic compounds in the plants. The content of chlorophyll in chloroplasts is a crucial factor affecting photosynthesis efficiency. Under stress conditions, the amount of this pigment can decrease, leading to a reduction in the rate of photosynthesis. Consequently, plant growth and development may be inhibited. Carotenoids,

which are synthesized in response to stress, play vital roles in various plant processes and act as potential antioxidants [77]. Finally, phenolic compounds serve as defense mechanisms against stressors. Phenolic can function as either antioxidants or pro-oxidant signals, influencing the production of secondary metabolites. They also act as signals during interactions between fungal pathogens and plants. Fungal pathogens can either metabolize phenolic or respond to them as signals, or in some cases, both [78].

In our experiments, the content of chlorophyll *a* varied among the tested cultivars, and was significantly lower in plants subjected to *R. solani* soil infestation compared to non-infested control. Chlorophyll *a* is the most common type of chlorophyll, present in all plants, algae, and cyanobacteria. It plays a vital role in transferring energy from sunlight to photosynthesis processes [79]. Variations in chlorophyll *a* content may serve as a compensatory mechanism for plants to offset the nutrient loss caused by *R. solani* infestation. Additionally, we observed statistical differences in chlorophyll *b* content between tested cultivars, as well as a significant increase in chlorophyll *b* content resulting from *R. solani* leaf infestation. Chlorophyll *b* is the second most important type of chlorophyll, primarily found in green plants (e.g., seed plants, ferns, and algae). It exhibits a higher capacity to absorb green light compared to chlorophyll *a*, complementing its light absorption and extending the range of light available for photosynthesis [80]. Interestingly, significant decreases were found in chlorophyll *a*/*b* ratios for leaves and soil infestations as compared to control (with leaf infestation presenting the lowest ratio). Moreover, differences were also found for chlorophyll *a*/*b* ratio between tested cultivars. The ratio of chlorophyll *a* to chlorophyll *b* provides insights into the health and functionality of plants. Changes in this ratio can indicate environmental stressors such as nutrient deficiency or insufficient light supply [81]. Furthermore, significant differences were also found in the chlorophyll (*a* +*b*)/carotenoids ratio among the tested cultivars. Interestingly, this ratio was highest for leaf infestation and did not differ between control and soil infestation. This ratio reflects the plant's overall ability to absorb and utilize light, with a direct impact on photosynthesis efficiency and energy production. Carotenoids, such as beta-carotene, lutein, and zeaxanthin, play a crucial role in photosynthesis by absorbing light in additional areas of the spectrum, particularly in the blue and violet range. Deviations from the normal carotenoid-to-chlorophyll ratio may signify pigment synthesis disorders or reactions to adverse environmental conditions. Evaluating these ratios aids in diagnosing and monitoring plant health [38].

Lastly, we found that the 'Kasia' cultivar exhibited the highest total phenolics content, while 'Luczniczka' the lowest. Phenolic compounds are widely distributed in plants and serve various functions. They are synthesized by plants in response to stress caused by infections, injuries, extreme temperatures, and UV radiation exposure. Phenolic can be found in different plant parts, including leaves, stems, roots, fruits, and seeds. These compounds play important roles in plant physiology, acting as antioxidants, protective agents against pathogens by affecting their enzymes or metabolic processes, regulators of growth and development, as well as the natural sunscreens that absorb UV radiation and protect plant tissues from its detrimental effects [82]. Since the highest phenolics biosynthesis occurred in plants cultivated in infested soil rather than in leaf-infested plants or in control plants, one can assume that *R. solani* has found in the soil a favorable environment for growth, triggering a defensive response in the plant.

## Conclusions

Our study investigated the effects of *Rhizoctonia solani* infection on the emission of volatile organic compounds (VOCs) and selected primary and secondary metabolites present in the leaves of ten chrysanthemum cultivars grown in a greenhouse. Our findings suggest that chrysanthemum cultivars exhibit distinct responses to *Rhizoctonia solani* infection, with some

cultivars emitting higher quantities of certain VOCs compared to others. Furthermore, the composition of VOCs emitted by infected plants varied over time, indicating dynamic changes in plant-pathogen interactions and metabolic responses.

These results represent a step forward in understanding the mechanisms of plant defense and susceptibility to fungal pathogens in chrysanthemum cultivars. By elucidating the specific VOCs involved in plant-pathogen interactions, future research can be focused on investigating the role of individual compounds in mediating plant defense mechanisms and pathogen virulence. Further studies could involve transcriptomic, proteomic, and metabolomic analyses to identify key genes, proteins, and metabolic pathways involved in VOC biosynthesis and regulation. The future prospects incorporate research that investigates the potential use of VOC profiles as diagnostic markers for early detection and monitoring of *Rhizoctonia solani* infection in chrysanthemum crops, which could help improve environmental-friendly disease management strategies and minimize crop losses.

## Supporting information

**S1 Table. *F*-statistics from two-way analysis of variance for observed VOCs.** (DOCX)

**S2 Table. Chlorophyll *a*/*b* and chlorophylls (*a*+*b*)/carotenoids ratio in leaves of ten chrysanthemum cultivars infested with *Rhizoctonia solani* on leaves and on the surface of growth substrate.** Mean values and standard deviations (s.d.) in columns and rows followed by the same letter do not differ significantly with two-way ANOVA and Fisher's post-hoc test at $p < 0.05$. Capital letters refer to the main effects (irrespectively), small letters refer to the interaction between the two studied independent variables. (DOCX)

**S3 Table. *F*-statistics from two-way analysis of variance for observed metabolites content in leaves.** (DOCX)

**S1 Fig. Mean values [ng hr$^{-1}$] for Z-3-HAL emission by *Chrysanthemum* × *morifolium* cultivars following the infestation of *Rhizoctonia solani*, collected on days 3 and 6 post-foliar application, and after soil-inoculation, respectively.** (DOCX)

**S2 Fig. Mean values [ng hr$^{-1}$] for E-2-HAL emission by *Chrysanthemum* × *morifolium* cultivars following the infestation of *Rhizoctonia solani*, collected on days 3 and 6 post-foliar application, and after soil-inoculation, respectively.** (DOCX)

**S3 Fig. Mean values [ng hr$^{-1}$] for Z-3-HOL emission by *Chrysanthemum* × *morifolium* cultivars following the infestation of *Rhizoctonia solani*, collected on days 3 and 6 post-foliar application, and after soil-inoculation, respectively.** (DOCX)

**S4 Fig. Mean values for [ng hr$^{-1}$] E-2-HOL emission by *Chrysanthemum* × *morifolium* cultivars following the infestation of *Rhizoctonia solani*, collected on days 3 and 6 post-foliar application, and after soil-inoculation, respectively.** (DOCX)

**S5 Fig. Mean values [ng hr$^{-1}$] for β-PIN emission by *Chrysanthemum* × *morifolium* cultivars following the infestation of *Rhizoctonia solani*, collected on days 3 and 6 post-foliar**

application, and after soil-inoculation, respectively.
(DOCX)

**S6 Fig. Mean values [ng hr⁻¹] for β-MYR emission by** *Chrysanthemum* × *morifolium* **cultivars following the infestation of** *Rhizoctonia solani***, collected on 3 and 6 post-foliar application, and after soil-inoculation, respectively.**
(DOCX)

**S7 Fig. Mean values [ng hr⁻¹] for Z-3-HAC emission by** *Chrysanthemum* × *morifolium* **cultivars following the infestation of** *Rhizoctonia solani***, collected on days 3 and 6 post-foliar application, and after soil-inoculation, respectively.**
(DOCX)

**S8 Fig. Mean values [ng hr⁻¹] for (Z)-OCI emission by** *Chrysanthemum* × *morifolium* **cultivars following the infestation of** *Rhizoctonia solani***, collected on days 3 and 6 post-foliar application, and after soil-inoculation, respectively.**
(DOCX)

**S9 Fig. Mean values [ng hr⁻¹] for BAC emission by** *Chrysanthemum* × *morifolium* **cultivars following the infestation of** *Rhizoctonia solani***, collected on days 3 and 6 post-foliar application, and after soil-inoculation, respectively.**
(DOCX)

**S10 Fig. Mean values [ng hr⁻¹] for LIN emission by** *Chrysanthemum* × *morifolium* **cultivars following the infestation of** *Rhizoctonia solani***, collected on days 3 and 6 post-foliar application, and after soil-inoculation, respectively.**
(DOCX)

**S11 Fig. Mean values [ng hr⁻¹] for MA emission by** *Chrysanthemum* × *morifolium* **cultivars following the infestation of** *Rhizoctonia solani***, collected on days 3 and 6 post-foliar application, and after soil-inoculation, respectively.**
(DOCX)

**S12 Fig. Mean values [ng hr⁻¹] for IND emission by** *Chrysanthemum* × *morifolium* **cultivars following the infestation of** *Rhizoctonia solani***, collected on days 3 and 6 post-foliar application, and after soil-inoculation, respectively.**
(DOCX)

**S13 Fig. Mean values [ng hr⁻¹] for β-CAR emission by** *Chrysanthemum* × *morifolium* **cultivars following the infestation of** *Rhizoctonia solani***, collected on days 3 and 6 post-foliar application, and after soil-inoculation, respectively.**
(DOCX)

**S14 Fig. Mean values [ng hr⁻¹] for β-FAR emission by** *Chrysanthemum* × *morifolium* **cultivars following the infestation of** *Rhizoctonia solani***, collected on days 3 and 6 post-foliar application, and after soil-inoculation, respectively.**
(DOCX)

## Acknowledgments

The authors wish to thank Dominika Rymarz and Anita Woźny from Bydgoszcz University of Science and Technology, Faculty of Agriculture and Biotechnology, for their technical support in greenhouse plant cultivation.

## Author Contributions

**Conceptualization:** Dariusz Piesik.

**Formal analysis:** Jan Bocianowski.

**Investigation:** Natalia Miler, Grzegorz Lemańczyk, Alicja Tymoszuk, Karol Lisiecki.

**Methodology:** Dariusz Piesik, Natalia Miler, Grzegorz Lemańczyk, Alicja Tymoszuk.

**Software:** Jan Bocianowski.

**Supervision:** Dariusz Piesik.

**Validation:** Krzysztof Krawczyk.

**Visualization:** Jan Bocianowski.

**Writing – original draft:** Natalia Miler, Grzegorz Lemańczyk, Alicja Tymoszuk, Krzysztof Krawczyk.

**Writing – review & editing:** Dariusz Piesik, Chris A. Mayhew.

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
