## [Decision Letter · Decision Letter 0]

24 Jan 2024

PONE-D-23-38974Induction of volatile organic compounds in chrysanthemums plants following infection by RhizoctoniaPLOS ONE

Dear Dr. Piesik,

Thank you for submitting your manuscript to PLOS ONE. After careful consideration, we feel that it has merit but does not fully meet PLOS ONE’s publication criteria as it currently stands. Therefore, we invite you to submit a revised version of the manuscript that addresses the points raised during the review process.

We look forward to receiving your revised manuscript.

Kind regards,

Muhammad Anwar, PHD

Academic Editor

PLOS ONE

Journal Requirements:

Reviewers' comments:

Reviewer's Responses to Questions

**Comments to the Author**

1. Is the manuscript technically sound, and do the data support the conclusions?

Reviewer #1: Partly

Reviewer #2: Yes

2. Has the statistical analysis been performed appropriately and rigorously? 

Reviewer #1: Yes

Reviewer #2: Yes

3. Have the authors made all data underlying the findings in their manuscript fully available?

Reviewer #1: Yes

Reviewer #2: No

4. Is the manuscript presented in an intelligible fashion and written in standard English?

Reviewer #1: Yes

Reviewer #2: Yes

5. Review Comments to the Author

Reviewer #1: This comprehensive study investigates the impact of Rhizoctonia solani infestation on ten cultivars of chrysanthemum, shedding light on both volatile organic compound (VOC) emissions and biochemical composition of plants. Through meticulous analysis using gas chromatography-mass spectrometry, the research reveals cultivar-specific variations in VOC emissions, with some displaying a broader spectrum compared to others. The investigation underscores the intricate interplay between cultivar, collection time, and Rhizoctonia infection, elucidating significant effects on VOCs. Notably, the study unveils alterations in biochemical composition induced by the infection, with a noteworthy decrease in carotenoid content and an increase in phenolics, offering valuable insights for disease management strategies and the development of resistant chrysanthemum cultivars. The manuscript is interesting and valuable, its novelty is clear. I only have some minor technical suggestions regarding the text.

- Please provide the initials of the authors when mentioning the species for the first time.

- Some older references in the Introduction can be updated.

- Some names of the cultivars are inconsistent throughout the text, e.g. Łuczniczka and Luczniczka.

- According to the Abstract, the plants were grown in controlled conditions in the glasshouse, yet there is no information on the light conditions.

- Avoid repetitions (see page 6).

- Sometimes in the text, you give detailed information about the producers of chemicals and equipment and sometimes only brief.

- Sometimes you use the abbreviation ml and sometimes mL (I suggest the latter one).

- Some parts of the text are unclear (see the corrected manuscript).

- Sometimes you are using a serial comma, and sometimes not.

- I suggest preparing a separate Conclusions chapter. What are the suggestions for future research?

- In the Reference list, Latin names should be written in italics.

- Grammar and punctuation require minor corrections.

For more specific comments, please see the corrected manuscript.

Reviewer #2: The manuscript is about the Induction of volatile organic compounds in chrysanthemum plants following Rhizoctonia solani infection. The manuscript contain important information for the scientific community. I have following suggestions to improve this.

Title: Change to “ chrysanthemum plants” instead of ”chrysanthemums plants” and add “solani” after Rhizoctonia as the study is on a specific bacterial infection.

Abstract;

The aim and the significance/application of the study was not well justified.

Introduction

Line 74- Please give some examples

Materials and methods

Line 134- What are the greenhouse conditions?

Line 145- Do you mean (Fig.1A)? Please be consistent.

Line 183- What is super-Q absorbent?

Line 204- Manufacturer of GC-MS?

Results:

Line 290 – “figures” and Line 299 “Figs”. Please be consistent.

Line 338- What trait?

Line 343- Please include the data not shown in supplementary materials to confirm significance.

6. PLOS authors have the option to publish the peer review history of their article (what does this mean?). If published, this will include your full peer review and any attached files.

Reviewer #1: No

Reviewer #2: No

---

## [Author Response · Author response to Decision Letter 0]

20 Feb 2024

Dear Reviewers,

We appreciate the opportunity to respond to the points raised in our recent publication, titled “Induction of volatile compounds in chrysanthemum plants following infection by Rhizoctonia”. After thorough review and analysis, we find the reviews as very helpful and constructive. We have addressed all the issues raised by the Reviewers. We hope that our answers will clarify all of them. Our answers, comments and corrections are listed below the Reviewers queries, typed in red.

Reviewer #1: This comprehensive study investigates the impact of Rhizoctonia solani infestation on ten cultivars of chrysanthemum, shedding light on both volatile organic compound (VOC) emissions and biochemical composition of plants. Through meticulous analysis using gas chromatography-mass spectrometry, the research reveals cultivar-specific variations in VOC emissions, with some displaying a broader spectrum compared to others. The investigation underscores the intricate interplay between cultivar, collection time, and Rhizoctonia infection, elucidating significant effects on VOCs. Notably, the study unveils alterations in biochemical composition induced by the infection, with a noteworthy decrease in carotenoid content and an increase in phenolics, offering valuable insights for disease management strategies and the development of resistant chrysanthemum cultivars. The manuscript is interesting and valuable, its novelty is clear. I only have some minor technical suggestions regarding the text.

Dear Reviewer #1, Thank you for your evaluation and supportive comments. We did our best to address them properly. Please, find our answers typed in red.

- Please provide the initials of the authors when mentioning the species for the first time.

It has been done.

- Some older references in the Introduction can be updated.

In the Introduction the following references (by numbers) have been updated: 10, 19, 22, 27 and 28

- Some names of the cultivars are inconsistent throughout the text, e.g. Łuczniczka and Luczniczka.

We have changed the typing of Łuczniczka to Luczniczka. Also, we have retyped Małgosia with Malgosia, following your suggestion and to keep consistency.

- According to the Abstract, the plants were grown in controlled conditions in the glasshouse, yet there is no information on the light conditions.

This information has been added to Material and methods. Also, in the Abstract, we added word “semi-” to “controlled” conditions, since the light provided to plants was from natural source.

- Avoid repetitions (see page 6).

It has been corrected.

- Sometimes in the text, you give detailed information about the producers of chemicals and equipment and sometimes only brief.

It has been corrected. All the manufacturers’ data have been adjusted to the name of the company and the country of production.

- Sometimes you use the abbreviation ml and sometimes mL (I suggest the latter one).

It has been corrected.

- Some parts of the text are unclear (see the corrected manuscript).

Thank you for all of your valuable comments in the manuscript, we have corrected all of them according to your suggestions. Thank you for your patience and time.

- Sometimes you are using a serial comma, and sometimes not.

It has been corrected, serial commas have been deleted.

- I suggest preparing a separate Conclusions chapter. What are the suggestions for future research?

The separate Conclusion chapter which presents also some future directions prospects has been added to the manuscript.

- In the Reference list, Latin names should be written in italics.

It has been corrected

- Grammar and punctuation require minor corrections.

It has been corrected

For more specific comments, please see the corrected manuscript.

We have followed all your suggestions.

Thank you for your time and effort, providing us with detailed, as well as essential comments, which have improved the quality of the manuscript. 

Reviewer #2: The manuscript is about the Induction of volatile organic compounds in chrysanthemum plants following Rhizoctonia solani infection. The manuscript contain important information for the scientific community. I have following suggestions to improve this.

Dear Reviewer#2, Thank you for your supportive comments. Please, find our answers typed in red below.

Title: Change to “chrysanthemum plants” instead of ”chrysanthemums plants” and add “solani” after Rhizoctonia as the study is on a specific bacterial infection.

It has been corrected.

Abstract;

The aim and the significance/application of the study was not well justified.

We have developed the aim and significance of our study in the Abstract, according to your suggestion. 

Introduction

Line 74- Please give some examples

The examples of certain GLVs action has been added to the sentence with new references (11, 12 and 13)

].

Materials and methods

Line 134- What are the greenhouse conditions?

We have added information concerning light and photoperiod conditions during the cultivation and infestation period. The temperature and humidity information had already been indicated in lines 144-146 of original manuscript.

Line 145- Do you mean (Fig.1A)? Please be consistent.

Thank you, it has been adjusted to keep consistency.

Line 183- What is super-Q absorbent?

Volatile Collection Traps (Super-Q), are passive chemical filters designed for the collection of extremely low-level (ppm-ppb) volatile organic compounds (VOC’s) from nalophan enclosed plants. This information has been added into the manuscript.

Line 204- Manufacturer of GC-MS?

The manufacturer of GC-MS was (Perkin Elmer, USA), it was indicated in line 208 of the original manuscript.

Results:

Line 290 – “figures” and Line 299 “Figs”. Please be consistent.

It has been corrected and adjusted in other places of manuscript.

Line 338- What trait?

It has been corrected. The sentence was improved with: “The mean chlorophyll a content in different cultivars regardless the Rhizoctonia treatments…”

Line 343- Please include the data not shown in supplementary materials to confirm significance.

The supplementary tables S2 and S3 has been included into Supplementary file.

---

## [Decision Letter · Decision Letter 1]

9 Apr 2024

Induction of volatile organic compounds in chrysanthemum plants following infection by Rhizoctonia solani

PONE-D-23-38974R1Dariusz Piesik

Dear Dr. Dariusz piesik,

We’re pleased to inform you that your manuscript has been judged scientifically suitable for publication and will be formally accepted for publication once it meets all outstanding technical requirements.

Kind regards,

Muhammad Anwar, PHD

Academic Editor

PLOS ONE

Additional Editor Comments (optional):

Reviewers' comments:

Reviewer's Responses to Questions

**Comments to the Author**

1. If the authors have adequately addressed your comments raised in a previous round of review and you feel that this manuscript is now acceptable for publication, you may indicate that here to bypass the “Comments to the Author” section, enter your conflict of interest statement in the “Confidential to Editor” section, and submit your "Accept" recommendation.

Reviewer #1: All comments have been addressed

Reviewer #3: All comments have been addressed

2. Is the manuscript technically sound, and do the data support the conclusions?

Reviewer #1: Yes

Reviewer #3: Yes

3. Has the statistical analysis been performed appropriately and rigorously? 

Reviewer #1: Yes

Reviewer #3: Yes

4. Have the authors made all data underlying the findings in their manuscript fully available?

Reviewer #1: Yes

Reviewer #3: Yes

5. Is the manuscript presented in an intelligible fashion and written in standard English?

Reviewer #1: Yes

Reviewer #3: Yes

6. Review Comments to the Author

Reviewer #1: (No Response)

Reviewer #3: This study reveals changes in the volatile and biochemical responses of chrysanthemum plants to R. solani infestation, which may contribute to the development of disease management strategies and improvement of chrysanthemum cultivars' resistance to R. solani. The article presents significant research that offers a new perspective on the interactions between chrysanthemum plants and the pathogen R. solani. The experiments were conducted meticulously, and the methods of analysis are appropriate for the stated research objectives. The results are clearly presented, facilitating the understanding of the conclusions. The finding that R. solani infestation affects VOC emissions and biochemical composition of chrysanthemum is significant for both scientific understanding and agricultural practice. The discovery of reduced carotenoid content due to infestation may have important implications for plant breeding and protection. Furthermore, the suggested application of the study's results in the development of resistant chrysanthemum cultivars represents a valuable contribution to the field of plant breeding.

Overall, this article deserves recognition as a significant contribution to the scientific literature in the fields of plant pathology and plant breeding.

7. PLOS authors have the option to publish the peer review history of their article (what does this mean?). If published, this will include your full peer review and any attached files.

Reviewer #1: No

Reviewer #3: No
